

# Remote sensing inversion and spatial variation of land surface temperature over mining areas of Jixi, Heilongjiang, China

Jia-shuo Cao[1,2], Zheng-yu Deng[1,2], Wen Li[1] and Yuan-dong Hu[1,2]

[1] College of Landscape Architecture, Northeast Forestry University, Harbin, Heilongjiang, China
[2] Key Laboratory for Garden Plant Germplasm Development & Landscape Eco-Restoration in Cold Regions of Heilongjiang Province, Harbin, Heilongjiang, China

## ABSTRACT

**Background.** Jixi is a typical mining city in China that has undergone dramatic changes in its land-use pattern of mining areas over the development of its coal resources. The impacts of coal mining activities have greatly affected the regional land surface temperature and ecological system.

**Methods.** The Landsat 8 Operational Land Imager (OLI) data from 2015 and 2019 were used from the Jiguan, Didao, and Chengzihe District of Jixi in Heilongjiang, China as the study area. The calculations to determine the land-use classification, vegetation coverage, and land surface temperature (LST) were performed using ArcGIS10.5 and ENVI 5.3 software packages. A correlation analysis revealed the impact of land-use type, vegetation coverage, and coal mining activities on LSTs.

**Results.** The results show significant spatial differentiation in the LSTs of Jixi City. The LSTs for various land-use types were ranked from high to low as follows: mining land > construction land > grassland > cultivated land > forest land > water area. The LST was lower in areas with high vegetation coverage than in other areas. For every 0.1 increase in vegetation coverage, the LST is expected to drop by approximately 0.75 °C. An analysis of mining land patches indicates that the patch area of mining lands has a significant positive correlation with both the average and maximum patch temperatures. The average patch temperature shows a logarithmic increase with the growth of the patch area, and within 200,000 $m^2$, the average patch temperature increases significantly. The maximum patch temperature shows a linear increase with the patch area growth, and for every 100,000 $m^2$ increase in the patch area of mining lands, the maximum patch temperature increases by approximately 0.81 °C. The higher the average patch temperature of mining land, the higher the temperature in its buffer zone, and the greater its influence scope. This study provides a useful reference for exploring the warming effects caused by coal mining activities and the definition of its influence scope.

Corresponding author
Wen Li, liwen@nefu.edu.cn

# INTRODUCTION

The land surface temperature (LST) comprehensively reflects the energy exchange between land and the atmosphere, which is an important geophysical parameter in the ground-air system (*Li et al., 2016*; *Zhu et al., 2016*). Coupling the inversion results of LST with other parameters, such as land-use type and vegetation coverage, provides a scientific basis for ecological environmental protection (*Li et al., 2014*; *Liang & Zhai, 2014*; *Xu, He & Huang, 2013*; *Zhang et al., 2013a*; *Zhang et al., 2013b*). The commonly used LST inversion algorithms are divided primarily into the single-channel algorithm, multi-channel algorithm, and split-window algorithm (*Zhu et al., 2016*). Among them, the single-channel algorithms include the atmospheric correction method, Mono-window algorithm, and the Jiménez-Muñoz J.C single-channel algorithm (*Qin, Karnieli & Berliner, 2001*; *Jiménez-Muñoz et al., 2008*). The multi-channel algorithms mainly include the day-night method, temperature emissivity separation algorithm, and graybody emissivity method (*Gan et al., 2006*; *Gillespie et al., 2002*; *Zhang et al., 2000*). The split window algorithm is based mostly on data from the Landsat-TIRS, NOAA-AVHRR, and TERRA-MODIS (*Rozenstein et al., 2014*; *Qin & Karnieli, 2001*; *Mao et al., 2005*).

Due to aggravation of the heat island effect, current research on LSTs is mostly focused on urban areas. Analyzing differences in LSTs for different land-use types optimizes the distribution of green space from the perspective of landscape patterning to reduce the heat island effect (*Liu, 2016*). However, mining areas, which are often affected by high temperatures and cause safety problems, have not attracted sufficient attention and are rarely studied.

Some research has shown that in the resource development process for resource-based cities, the land-use patterns in mining areas are constantly changing, which causes a series of impacts on the regional ecological environment ( *Li et al., 2018*; *Chabukdhara & Singh, 2016*; *Xie, Wang & Fu, 2011*). Therefore, research focusing on coupling between land-use patterns in mining areas and the ecological environment indicators, such as the LST, water environment quality, and biodiversity, has become vital to environmental sustainability (*Zhou & Wang, 2014*; *Xiao, Hu & Fu, 2014*; *Hu, Duo & Wang, 2018*; *Bian et al., 2018*). Current research on land surface temperatures in mining areas mainly includes the temporal and spatial distribution characteristics of the surface temperature, the impact of ecological disturbance on the surface temperature, and others, where the scales are mostly at macro-regions (*Li, Yang & Lei, 2017*; *Qiu & Hou, 2013*; *Xie, Wang & Fu, 2011*). This study specifically analyzes the overall and local distribution characteristics of LSTs from smaller scales to explore the radius of influence of high-temperature points. This provides a reference to establish heat alerts in mining areas.

Jixi is a typical mining city in China that has undergone dramatic changes in its land-use pattern in the mining area during the development of coal resources. Significant amounts of cultivated land, forest land, and other land types have been replaced by industrial and mining sites, which has greatly affected the regional ecological environment. Impacts such as the atmospheric diffusion of pollutants and the rise of LSTs have affected the regional landscape and ecological systems (*Pan et al., 2013*; *Liao, 2009*). This paper uses data from

the Landsat 8 OLI remote sensing images from 2015 and 2019 to determine LSTs using the radiation conduction equation over the study area, which encompasses the Jiguan, Didao, and Chengzihe District of Jixi. We analyzed the spatial differentiation and correlations of the LST with the land-use type and vegetation coverage to provide a theoretical framework to reduce the heat island effects caused by local urban development and coal mining activities.

## MATERIALS & METHODS

### Overview of the study area

The study area encompasses the Jiguan, Didao, and Chengzihe District of Jixi, which are the main mining lands with a total area of 827.87 km$^2$. Jixi is located in the southeast of Heilongjiang Province, between 130°24′24″–°56′30″E, 44°51′12″–46°36′55″N. To the southeast and across the ocean in Russia, while to the west and south are Mudanjiang, and to the north is Qitaihe (Fig. 1). The province comprises Mishan, Hulin, and Jidong Counties and six other districts (Jiguan, Hengshan, Didao, Chengzihe, Lishu, and Mashan). The study area is part of the cold-temperate continental monsoon climate, where the average annual temperature is 3.7 °C, the average precipitation is 537.5 mm, the annual sunshine is 2709 h and the average frost-free period is 140 d. The terrain is composed primarily of mountains, hills and plains.

Jixi is relatively rich in mineral resources with mutiple mining areas. However, there also are several abandoned mines that severely damage the ecological environment. In addition, urban construction and industrial development have encroached on grasslands, woodlands, and wetlands, which increases the ecological vulnerability and risks in these ecosystems (He, 2010).

### Data sources and preprocessing treatments

This paper is based on the Landsat 8 OLI remote sensing images from 2015 and 2019, all of which are from the US Geological Survey (http://glovis.usgs.gov/). All images have a spatial resolution of 30 m. The image strip numbers/rows used in this study are 115/28 and 115/29, respectively, and the imaging time was from July to September. Cloud cover in these images was less than 2%, and they were interpreted and classified based on a series of preprocessing treatments, including radiation calibration, atmospheric correction, band synthesis and image cropping.

### Analytical methods

The spatial differentiation characteristics of the LST in the Jiguan, Didao, and Chengzihe Districts of Jixi were used to identify heat islands and their influencing factors. We selected a single-window algorithm for inversion of the LSTs. These results were used to analyze the effects of the land-use type, vegetation coverage and coal mining activities on the spatial distribution of LSTs.

### Determining land-use classification and vegetation coverage

Land use is the most direct manifestation of the interaction between human activities and the natural environment as it reflects this close relationship in both time and space

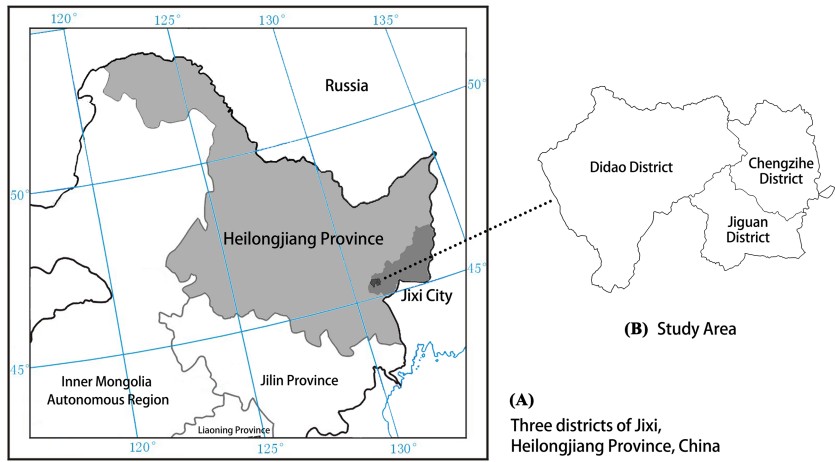

**Figure 1** **Location map showing the three districts of Jixi comprising the study area.** Map representing the geostrategic importance of the study area: (A) Jixi City, Heilongjiang Province, China; (B) three districts of Jixi.

(*Mooney, Duraiappah & Larigauderie, 2013*; *Liu et al., 2014*). Typically, areas designated as land resources reflects the status of natural resources within the study area. Changes in land-use patterns inevitably cause changes in the LSTs and ecosystem functionality. Therefore, the study of land use is of great importance for regional ecological analyses (*Marceau et al., 2003*).

The relationship between vegetation coverage and the LST has become a focus of research on heat islands (*Wang et al., 2011*). Green vegetation affects LSTs through photosynthesis, transpiration and evapotranspiration. *Ma et al. (2010)* compared and analyzed five correlation degrees among planting parameters and LSTs, including the normalized difference vegetation index (NDVI), ratio vegetation index (RVI), greenness vegetation index (GVI), modified soil to adjust vegetation index (MSAVI) and vegetation coverage. They concluded that the correlation between vegetation coverage and the LST was both high and stable because it is not markedly influenced by spatial location or changes in the fraction or type of surface coverage. Therefore, the relationship between vegetation coverage and the LST was selected to study heat island effects within different land surfaces.

## Land-use classification

The ENVI 5.3.1 (L3Harris Geospatial Solutions, Inc., Melbourne, FL, USA) and ArcGIS 10.5 (Esri, Corp., Redlands, CA, USA) were used to preprocess the original image data, which includes geometric correction, mosaic compilation, fusion, clipping, research scope extraction, image enhancement and supervised classification, before interpreting and analyzing the remote sensing imagery. The classification of land-use types in the study area was consistent with the standard land-use classification (GB/T 21010–2017). The study area was divided into six categories: forest lands, grasslands, construction lands, cultivated lands, mining lands and water areas. A maximum-likelihood approach was used for the

classification. In the final stage of the study, the remote sensing image interpretation was validated by site surveys. The accuracy of the results was verified by establishing a confusion matrix. Random points were selected in the Erdas Imagine 2015 software for classification, where a certain number of random points were selected for each category. The classification of each random point was distinguished visually so that the category to which each random point belongs is defined in the software. The user accuracy, producer accuracy, and Kappa coefficient of the overall classification of each category were then calculated.

## Vegetation coverage calculation

Plant coverage information is typically extracted from remote sensing images. Given the high accuracy of NDVI values estimated using remote sensing, it is one of the most widely used indexes (*Mu et al., 2012*). A common method to calculate vegetation coverage is based on the hybrid pixel decomposition method, where it is assumed that each pixel of the remote sensing image is composed of soil and vegetation components. Thus, the information includes both a pure soil component and a pure vegetation component. In this case, we assumed that the NDVI value is a weighted average sum of the index values from both soil and vegetation information (*Li, Fan & Wang, 2010*), which is given as follows:

$$NDVI = f_v \times NDVI_{veg} + (1 - f_v) \times NDVI_{soil}, \tag{1}$$

Where NDVI is the vegetation index value of mixed pixels; $NDVI_{veg}$ is the vegetation index of pure vegetation pixels; $NDVI_{soil}$ is the vegetation index value of pure soil pixels; and $f_v$ is the vegetation coverage. Thus, the formula for vegetation coverage ($f_v$) becomes:

$$f_v = (NDVI - NDVI_{soil}) / (NDVI_{veg} - NDVI_{soil}). \tag{2}$$

In practice, the parameters can be selected in the following ways. (1) Take different $NDVI_{veg}$ and $NDVI_{soil}$ values for different soil and vegetation types. (2) Use the maximum and minimum NDVIs of the study area, $NDVI_{veg} = NDVI_{max}$, $NDVI_{soil} = NDVI_{min}$. (3) Determine the NDVI value of the corresponding pixel based on measured data (*Li et al., 2015*). Under the influence of varying meteorological conditions, vegetation type and distribution, seasons, and other factors, both the $NDVI_{soil}$ and $NDVI_{veg}$ values for different images vary to some extent.

The maximum and minimum values of the given confidence interval are selected, and the confidence value is determined primarily from the image size and clarity. As a comparison, the maximum NDVI images of 2015 and 2019 were extracted. In the NDVI frequency accumulation table, the NDVI with a frequency of 5% was selected for $NDVI_{soil}$, and the NDVI with a frequency of 95% was selected for $NDVI_{veg}$. Finally, the vegetation coverage was obtained from Eq. (2).

## Land surface temperature inversion

The LST inversion algorithms for single-infrared-band Landsat 8 OLI remote sensing data are based primary on the radioactive transfer equation (RTE), a universal single-channel algorithm, and a single-window algorithm (*Ding & Xu, 2008*). Therefore, the RTE was selected to invert the LSTs in this study.

## Calculation of specific surface emissivity

Remote sensing images were firstly classified into three types: water bodies, towns and natural surfaces. The specific emissivity of water pixels is 0.995, where other surface emissivity estimates were based on the following formulas (*Chi, Zeng & Wang, 2016*):

$$\varepsilon_{\text{surface}} = 0.9625 + 0.0614 f_v - 0.0461 f_v^2 \tag{3}$$

$$\varepsilon_{\text{building}} = 0.9589 + 0.086 f_v - 0.0671 f_v^2 \tag{4}$$

Where $\varepsilon_{\text{surface}}$ and $\varepsilon_{\text{building}}$ represent the specific emissivity of natural surface pixels and urban pixels, respectively.

## Radioactive transfer equation

The RTE is also called the atmospheric correction method. It firstly estimates the impact of the atmosphere on the surface thermal radiation based on the information received by the satellite thermal infrared sensor. This is then subtracted from the total thermal radiation obtained by the sensor. The impact of the atmosphere on the surface can be used to obtain the intensity of surface thermal radiation. Assuming that the surface and the atmosphere have Lambertian properties for thermal radiation, the corresponding LST can be obtained as (*You & Yan, 2009*; *Yue et al., 2019*):

$$L_\lambda = \left[ \varepsilon \cdot B(T_S) + (1 - \varepsilon) L_\downarrow \right] \cdot \tau + L_\uparrow, \tag{5}$$

Where $L_\lambda$ is the intensity of thermal radiation received by the satellite sensor, $\varepsilon(K)$ is the surface emissivity, $T_S$ is the true LST, $B(T_S)$ (W m$^{-2}$ sr$^{-1}$ μm$^{-1}$) is the black body brightness corresponding to temperature $T_S$ derived from Planck's law, $\tau$ is the transmittance of the atmosphere at thermal infrared wavelengths, $L_\uparrow$ (W m$^{-2}$ sr$^{-1}$ μm$^{-1}$) is the atmospheric upward radiance, and $L_\downarrow$ (W m$^{-2}$ sr$^{-1}$ μm$^{-1}$) is the atmospheric downward radiance. Based on the RTE, the $B(T_S)$ can be obtained as (*Wu et al., 2016*; *Hou & Zhang, 2019*):

$$B(T_S) = \left[ L_\lambda - L_\uparrow - \tau \cdot (1 - \varepsilon) L_\downarrow \right] / \tau \cdot \varepsilon, \tag{6}$$

Where $\tau$, $L_\uparrow$ (W m$^{-2}$ sr$^{-1}$ μm$^{-1}$) and $L_\downarrow$ (W m$^{-2}$ sr$^{-1}$ μm$^{-1}$) were determined from the official NASA website (http://atmcorr.gsfc.nasa.gov/) by inputting the imaging time, latitude and longitude, air pressure and other relevant information to the study area. After estimating the of black body radiance $B(T_S)$, which is the same as the real temperature on the ground, the inverse function of Planck's law gives the real temperature on the ground as (*Chen, 2014*):

$$T_S = K_2 / ln \left( \frac{K_1}{B(T_S)} + 1 \right), \tag{7}$$

Where $K_1$ and $K_2$ are constants obtained by querying the Landsat metadata file. In this case, $K_1 = 774.8853$ and $K_2 = 1321.0789$ for Landsat 8 TIRS band 10.

## Normalized temperature index and temperature classification

The ecological environment of coal mining areas is damaged to varying degrees, this changes their LSTs and causes a series of significant ecological effects and environmental problems, such as vegetation degradation and soil erosion (*Dutta & Agrawal, 2003*; *Zhou & Zhang, 2005*). We used the urban heat island effect to explore the impact of coal mining activities on LSTs (*Ye et al., 2011*; *Li et al., 2019*). The formula for the normalized temperature index is:

$$T_r = \frac{\Delta T}{T_{\text{range}}} = \frac{T - T_{\text{min}}}{T_{\text{max}} - T_{\text{min}}}, \tag{8}$$

Where $T_r$ is the normalized temperature index, $T$ is the temperature at any spatial position in the region, $T_{\text{max}}$ and $T_{\text{min}}$ are the highest and lowest temperature in the region, respectively.

The method of equal intervals is used to divide the temperature based on the site conditions and existing research (*Sheng et al., 2010*; *Jia & Liu, 2006*). Once the maximum and minimum values of the inversion temperature are taken as endpoints, the temperature is divided into five equal-spaced intervals. These are a low-temperature zone, a low-middle-temperature zone, a middle-temperature zone, a middle-high-temperature zone, and a high-temperature zone. The normalized temperature indices for these levels were 0.0–0.2, 0.2–0.4, 0.4–0.6, 0.6–0.8, and 0.8–1.0, respectively (Table 1). Analyzing changes in the LST index at different distances from the mine allows evaluating the intensity and range of the heat island effect as caused by coal-mining activities.

## Analytical method of factors affecting land surface temperature

The terrain over the study area is relatively flat, which facilitates farming, town construction, and coal mining activities. We analyzed the spatial differentiation of LSTs in this area, which was linked to land use, vegetation coverage and coal mining activities.

## The influence of land-use classification on land surface temperature

The area and proportion of different types of land use were counted separately. Subsequently, the land-use and the LST maps were superimposed to obtain statistical data on the LSTs of various land-use types.

## The influence of vegetation coverage on land surface temperature

A profile analysis more intuitively reflected the relationship between changes in LST and vegetation coverage at a given geographical location. Using the interpolation line function in ArcGIS 10.5 to view profile values of LST and vegetation coverage from 2015 and 2019 to compare and analyze their associated changes along profiles to evaluate the relationships between these variables.

## The influence of coal mining activities on land surface temperature
### *The influence of patch area*

Firstly, all mining areas within a distance of 1,500 m from the edge of the study area were screened. These selected mining area patches were then counted and grouped based on area. We then combined these data with our LST inversion to determine the maximum,
**Table 1** The relationship between the normalized temperature index values and assigned temperature grades.

| Normalized temperature index | Temperature grade |
| --- | --- |
| 0.0–0.2 | Low temperature zone |
| 0.2–0.4 | Low-middle temperature zone |
| 0.4–0.6 | Middle temperature zone |
| 0.6–0.8 | Middle-high temperature zone |
| 0.8–1.0 | High temperature zone |

minimum, and average LSTs for different patches. Finally, the influence of these mining land patches on the LSTs were evaluated.

### The influence of buffer range

Buffers with a range of 100–1,500 m at intervals of 100 m were set for each of the patches. The average LST in each buffer zone was extracted, and the trends in the LSTs at varying distances from the mining area were analyzed.

## RESULTS

### Land surface temperature inversion

The LST results for the Jiguan, Didao and Chengzihe Districts of Jixi in 2015 and 2019 are shown in Fig. 2 and Table 2. The temperatures in 2015 were in general higher than those in 2019. The average LST over the entire study area was 25.64 °C in 2015 and 22.10 °C in 2019. There is a similarity in the spatial distribution patterns of their LSTs. High temperatures are concentrated in the south-central and southeast parts of the study area, while the temperatures in the west and north are relatively low. In these two years, the average LST in the Jiguan District was higher than averages in the other two districts, but its highest temperature was lower than the maximum recorded in the Didao and Chengzihe Districts. The highest temperatures over the entire study area were 42.29 °C, which was recorded at Shenghe Coal Mine in the Didao District. Likewise, the highest temperature in the Chengzihe District was recorded at Chengshan Coal Mine. Thus, mining areas had much higher LSTs than average. While only two years were selected for the analysis, similar results validate the conclusions.

The LSTs from 2015 and 2019 were normalized and divided into five levels, as shown in Fig. 3 and Table 3. The LSTs in the study area were assigned primarily to the low-temperature, low-middle-temperature, and middle-temperature zones, which covered the LST range of 19.16–33.04 °C in 2015 and 16.29–29.37 °C in 2019. Among them, the low-middle-temperature zone had the largest area as it accounted for more than 70% of the total study area. The high-temperature and middle-high-temperature zones had smaller areas. The high-temperature zone was distributed primarily within the Didao and Chengzihe Districts. The Shenghe Coal Mine accounted for 53.08% of the total area of the high-temperature zone in 2015 and rose to 59.04% in 2019. The proportion of the Chengshan Coal Mine in the total area of the high-temperature zone increased from

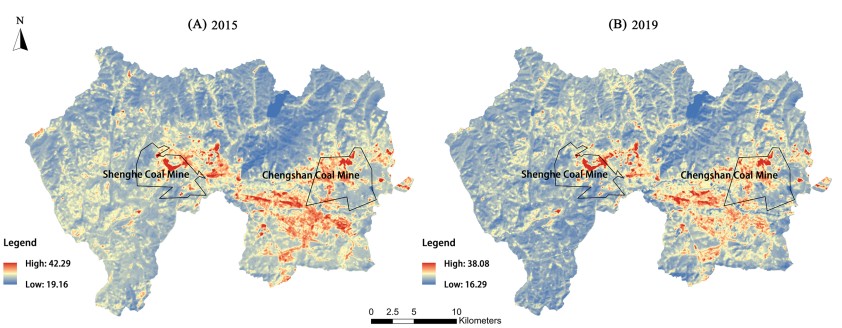

**Figure 2** **Land surface temperature (°C) results for the three districts of Jixi in 2019.** Land surface temperature (LST) maps for (A) 2015, (B) 2019 of the three districts in Jixi, Heilongjiang, China.

**Table 2** **Statistics on LST for the study area in 2015 and 2019.**

| Range | Land surface temperature/°C | | | | | | | |
|---|---|---|---|---|---|---|---|---|
| | 2015 | | | | 2019 | | | |
| | MEAN | MIN | MAX | STD | MEAN | MIN | MAX | STD |
| Jiguan District | 27.16 | 21.58 | 38.97 | 2.52 | 23.24 | 17.42 | 33.64 | 2.23 |
| Didao District | 25.23 | 19.16 | 42.29 | 1.92 | 21.75 | 17.18 | 38.08 | 1.63 |
| Chengzihe District | 25.53 | 19.45 | 39.13 | 2.48 | 22.34 | 16.29 | 35.26 | 2.14 |
| Total | 25.64 | 19.16 | 42.29 | 2.28 | 22.10 | 16.29 | 38.08 | 1.95 |

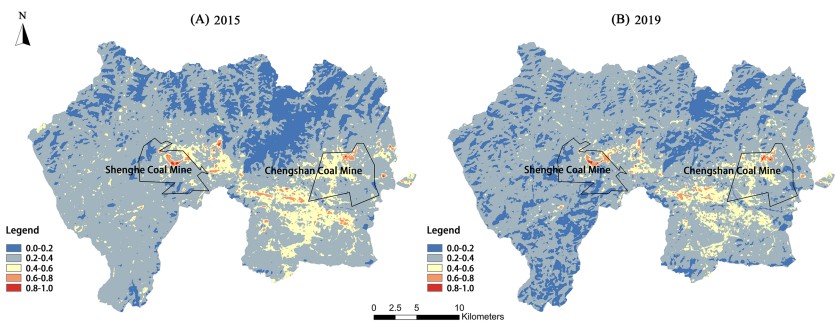

**Figure 3** **Spatial distribution of land surface temperature levels of the study area in 2015 and 2019.** Spatial distribution of land surface temperature levels for (A) 2015, (B) 2019 of the three districts in Jixi, Heilongjiang, China.

8.17% to 34.47% over these four years. Meanwhile, the low-temperature and low-middle-temperature zones were distributed mostly in the Didao and Chengzihe Districts, giving a large temperature difference between them. Therefore, local heat island effects were obvious within the study area.

## Land-use classification

Land-use types in the Jiguan, Didao, and Chengzihe Districts of Jixi in 2015 and 2019 are shown in Fig. 4 and Table 4. From 2015 to 2019, the area of forest land increased while

**Table 3  LST normalization results for the study area in 2015 and 2019.**

| Temperature grade | Normalized temperature index | 2015 | | 2019 | |
|---|---|---|---|---|---|
| | | LST /°C | Percentage | LST /°C | Percentage |
| Low temperature zone | 0.0–0.2 | 19.16–23.78 | 18.19% | 16.29–20.65 | 19.31% |
| Low-middle temperature zone | 0.2–0.4 | 23.78–28.41 | 70.53% | 20.65–25.01 | 72.21% |
| Middle temperature zone | 0.4–0.6 | 28.41–33.04 | 10.34% | 25.01–29.37 | 7.79% |
| Middle-high temperature zone | 0.6–0.8 | 33.04–37.66 | 0.90% | 29.37–33.72 | 0.66% |
| High temperature zone | 0.8–1.0 | 37.66–42.29 | 0.04% | 33.72–38.08 | 0.03% |
| Total | 0.0–1.0 | 19.16–42.29 | 100.00% | 16.29–38.08 | 100.00% |

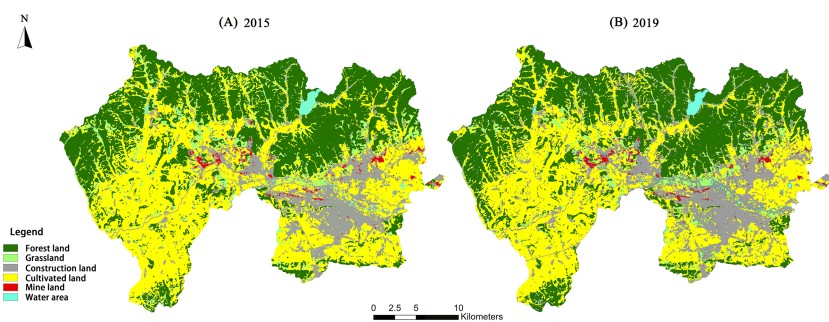

**Figure 4  Land-use types of the study area in 2015 and 2019.** Land-use types for (A) 2015, (B) 2019 of the three districts in Jixi, Heilongjiang, China.

the area of cultivated land decreased. However, the dominant land-use types in the study area are still forest land and cultivated land. The forest land is distributed mostly in the northern part of the study area, while the cultivated land is distributed in the middle and southern parts. Construction land is concentrated in the Jiguan District, which increased significantly from 109.94 km$^2$ to 133.69 km$^2$ in the four considered years. The mining land is defined primarily by the Shenghe Coal Mine in the Didao District and the Chengshan Coal Mine in the Chengzihe District. The accuracy of the land-use classification was verified by establishing a confusion matrix. The matrix showed that the Kappa coefficients of the land-use maps in the interpreted periods are all above 0.8, which meets the accuracy requirements for this study (Table 5).

## Vegetation coverage

The remote sensing images of the study area were processed according to the mixed pixel decomposition method to obtain the vegetation coverage of the Jiguan, Didao, and Chengzihe District of Jixi (Fig. 5). The construction land in the eastern Jiguan District, Shenghe Coal Mine in the Didao District and Chengshan Coal Mine in the Chengzihe District had the lowest vegetation coverage. However, ongoing urbanization and coal mining activities have markedly affected vegetation coverage in many other areas as well.

**Table 4  Land-use structure for the study area in 2015 and 2019.**

| Land-use | 2015 | | 2019 | |
|---|---|---|---|---|
| | Area / km² | Percentage / % | Area / km² | Percentage / % |
| Forest land | 294.07 | 35.52% | 304.18 | 36.74% |
| Grassland | 52.95 | 6.39% | 80.4 | 9.71% |
| Construction land | 109.94 | 13.28% | 133.69 | 16.15% |
| Cultivated land | 357.39 | 43.17% | 295.07 | 35.64% |
| Mining land | 7.10 | 0.86% | 6.76 | 0.82% |
| Water area | 6.42 | 0.78% | 7.77 | 0.94% |
| Total | 827.87 | 100.00% | 827.87 | 100.00% |

**Table 5  Accuracy evaluation of land use classification for the study area in 2015 and 2019.**

| Land-use | 2015 | | | | | | |
|---|---|---|---|---|---|---|---|
| | Forest land | Grassland | Construction land | Cultivated land | Mining land | Water area | Total |
| Forest land | 1,646 | 2 | – | – | – | – | 1,648 |
| Grassland | – | 150 | – | – | – | – | 150 |
| Construction land | 3 | – | 2406 | – | – | – | 2,409 |
| Cultivated land | – | 4 | 2 | 1737 | – | – | 1,743 |
| Mining land | – | – | 17 | – | 346 | – | 363 |
| Water area | – | – | – | – | – | 319 | 319 |
| Total | 1649 | 156 | 2425 | 1737 | 346 | 319 | 6,632 |
| Producers Accuracy | 99.82 | 96.15 | 99.22 | 1000 | 100 | 98.46 | – |
| Users Accuracy | 99.88 | 100 | 99.67 | 99.66 | 95.32 | 100 | – |
| | **2019** | | | | | | |
| Forest land | 858 | 35 | – | – | – | – | 893 |
| Grassland | – | 37 | 34 | 10 | 1 | – | 82 |
| Construction land | – | – | 1710 | 35 | 18 | – | 1,763 |
| Cultivated land | 2 | 11 | 19 | 821 | 2 | – | 855 |
| Mining land | – | – | 2 | – | 263 | – | 265 |
| Water area | – | – | – | – | – | 231 | 231 |
| Total | 860 | 83 | 1765 | 866 | 284 | 231 | 4,089 |
| Producers Accuracy | 99.77 | 44.58 | 96.88 | 94.8 | 92.61 | 92.4 | – |
| Users Accuracy | 96.08 | 45.12 | 96.07 | 95.8 | 99.25 | 100 | – |

**Notes.**
In 2015, Overall Classification Accuracy = 99.50%; Overall Kappa Statistics = 0.9932; In 2019, Overall Classification Accuracy = 95.42%; Overall Kappa Statistics = 0.9361.

## Correlation between land surface temperature and land-use types

The main land types in the low-temperature and low-middle-temperature zone are water areas, forest land, grassland and cultivated land. The main land types in the high-temperature, middle-high-temperature, and middle-temperature zones are construction land and mining land. There are large difference in the average LSTs among these land-use types (Table 6). The average LSTs for mining land, construction land and grassland were higher than the average LST for the study area. Among them, mining land had the highest average LSTs (33.33 °C in 2015 and 29.63 °C in 2019), yielding temperature anomalies of
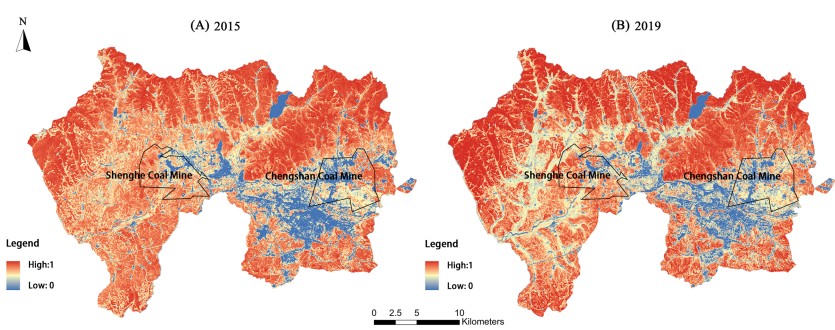

**Figure 5** **Vegetation coverage of the study area in 2015 and 2019.** Vegetation coverage for (A) 2015, (B) 2019 of the three districts in Jixi, Heilongjiang, China.

**Table 6** **Statistics on LST of different land-use types in 2015 and 2019.**

| Land use types | Land surface temperature/°C | | | | | | | |
|---|---|---|---|---|---|---|---|---|
| | 2015 | | | | 2019 | | | |
| | MEAN | MIN | MAX | STD | MEAN | MIN | MAX | STD |
| Forest land | 23.95 | 20.79 | 30.89 | 1.01 | 21.07 | 17.38 | 26.41 | 0.97 |
| Grassland | 26.55 | 21.94 | 36.27 | 1.45 | 23.21 | 18.77 | 30.56 | 1.43 |
| Construction land | 29.12 | 20.59 | 41.74 | 2.26 | 24.62 | 17.24 | 35.04 | 1.96 |
| Cultivated land | 25.74 | 21.35 | 33.71 | 1.20 | 21.73 | 18.26 | 29.14 | 1.09 |
| Mining land | 33.33 | 24.27 | 42.29 | 2.50 | 29.63 | 20.35 | 38.08 | 2.31 |
| Water area | 21.72 | 19.16 | 29.12 | 2.30 | 19.31 | 16.29 | 27.56 | 1.74 |

7.69 °C in 2015 and 7.53 °C in 2019. The water area had the lowest average LSTs (21.72 °C in 2015 and 19.31 °C in 2019). At the same time, the temperature standard deviation within the mining land was also relatively large, with a difference of 18.02 °C between the minimum and maximum temperatures.

## Correlation between land surface temperature and vegetation coverage

An east–west transect was drawn across the study are, and the data from 2019 were used to analyze changes in the LSTs with vegetation coverage. Every 25 pixel points on the profile were assigned to a group, and the average value of the vegetation coverage and LST in each group was calculated to obtain 56 data sets. Finally, a linear fit was performed between the vegetation coverage and average LST, and the coefficient of determination was assessed (Fig. 6). Areas with low vegetation coverage were associated with higher LSTs. In addition, as vegetation coverage decreased, the LSTs increased. The trends in LST and vegetation coverage were opposite with reciprocal change patterns.

The linear fit of the average LST and vegetation coverage (Fig. 7) shows that if the vegetation coverage increases by 0.1, the average LST is expected to decrease by approximately 0.75 °C. This constitutes a strong negative relationship between the LST and vegetation coverage. Using the SPSS 24 (IBM, Corp., Armonk, NY, USA) indicated a

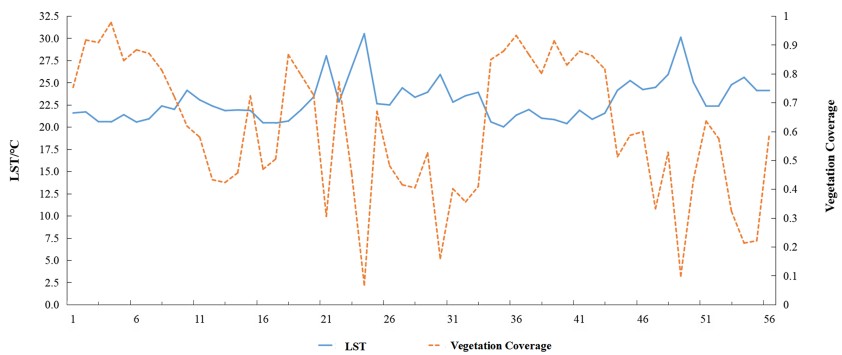

**Figure 6** Variation in land surface temperature (LST) and vegetation coverage in pixel groups (1–56) along an E-W profile.

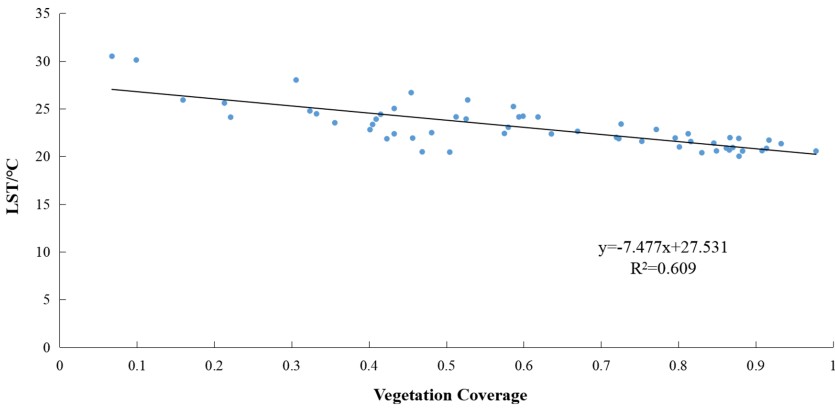

**Figure 7** Correlation between land surface temperature (LST) and vegetation coverage of the study area.

correlation coefficient of $R = -0.780$. This indicates a significant correlation at the 0.01 confidence level (both sides). Thus, green vegetation has a significant cooling effect on the land surface.

## Correlation between land surface temperature and coal mining activities

This study mainly considers spatial variations when exploring the correlation between the LST and mining activities. Therefore, the data of the most recent year (2019) is selected for the analysis, and the spatial distribution of the LST is analyzed based on the patch area and buffer sizes.

## Correlation between land surface temperature and patch area of mining lands

The mining areas were grouped based on patch area after screening them within 1500 m of the edge of the study area. The maximum, minimum and average LSTs of each patch were calculated from the 52 data sets (Table 7). Correlations among the average patch area and

**Table 7  Statistics on LST and patch area for mining lands.**

| Average area /m² | LST/°C | | Average area /m² | LST/°C | | Average area /m² | LST/°C | |
|---|---|---|---|---|---|---|---|---|
| | MEAN | MAX | | MEAN | MAX | | MEAN | MAX |
| 900 | 26.88 | 30.98 | 18000 | 29.09 | 30.27 | 53100 | 30.22 | 31.95 |
| 1800 | 27.38 | 32.30 | 19800 | 29.17 | 35.11 | 58500 | 27.85 | 29.48 |
| 2700 | 27.25 | 30.87 | 20700 | 31.85 | 33.43 | 61200 | 29.79 | 31.73 |
| 3600 | 26.50 | 30.49 | 21600 | 29.60 | 30.61 | 65700 | 31.90 | 34.51 |
| 4500 | 27.97 | 31.39 | 23400 | 29.29 | 30.25 | 82800 | 30.79 | 32.06 |
| 5400 | 28.23 | 30.16 | 24300 | 28.47 | 31.40 | 88200 | 29.63 | 31.78 |
| 6300 | 27.46 | 31.61 | 25200 | 29.02 | 30.51 | 110700 | 28.62 | 30.18 |
| 7200 | 27.26 | 30.87 | 26100 | 27.89 | 29.30 | 114300 | 31.36 | 33.22 |
| 8100 | 26.52 | 29.54 | 27000 | 28.26 | 29.43 | 135000 | 28.94 | 31.05 |
| 9000 | 27.57 | 29.63 | 28800 | 28.59 | 31.25 | 139500 | 28.70 | 31.22 |
| 9900 | 28.39 | 30.63 | 31500 | 27.54 | 30.19 | 162000 | 29.85 | 32.49 |
| 10800 | 28.23 | 31.98 | 32400 | 28.33 | 31.92 | 175500 | 30.14 | 32.31 |
| 12600 | 28.26 | 33.02 | 36000 | 28.48 | 30.26 | 179100 | 30.13 | 32.35 |
| 13500 | 29.44 | 30.58 | 39600 | 28.54 | 30.53 | 241200 | 29.04 | 31.41 |
| 14400 | 26.99 | 29.11 | 40500 | 27.86 | 30.36 | 490500 | 31.15 | 33.10 |
| 15300 | 25.58 | 26.63 | 43200 | 28.33 | 30.66 | 626400 | 31.58 | 35.26 |
| 16200 | 27.53 | 28.46 | 48600 | 29.85 | 31.84 | 754200 | 31.58 | 38.08 |
| 17100 | 26.57 | 29.39 | | | | | | |

the average and maximum patch temperatures were analyzed using SPSS 24. Our analysis indicates that the patch was strongly positively correlated with the average and maximum patch temperatures.

Correlation between the patch area and average patch temperature (Fig. 8) yielded $R = 0.571$. This indicates a significant correlation at the 0.01 confidence level (both sides). The determination coefficient of the fit logarithmic function was $R^2 = 0.487$, indicating that larger patch sizes promote a greater average patch temperature. Within 200,000 m², the average patch temperature increases rapidly with the size of the patch area. Once above 200,000 m², the average patch temperature increases more slowly.

The correlation between the patch area and maximum patch temperature (Fig. 9) yielded $R = 0.645$. This indicates a significant correlation at the 0.01 confidence level (both sides). The determination coefficient of the linear fit was $R^2 = 0.415$, indicating that larger patch sizes promote a greater maximum patch temperature. If the patch area of mining land increases by 100,000 m², the maximum patch temperature will increase by approximately 0.81 °C.

### Correlation between land surface temperature and various buffer sizes

The schematic diagram of buffer zone in mining land patch is shown in Fig. 10. A correlation analysis was performed on average patch area, average patch temperature, maximum patch temperature of mining land and the average LST in buffer zones at 100–1,500 m reviewed at 100 m intervals (Table 8). The temperature of the buffer zones within 0–100 m was strongly correlated with the patch area, average patch temperature, and maximum patch

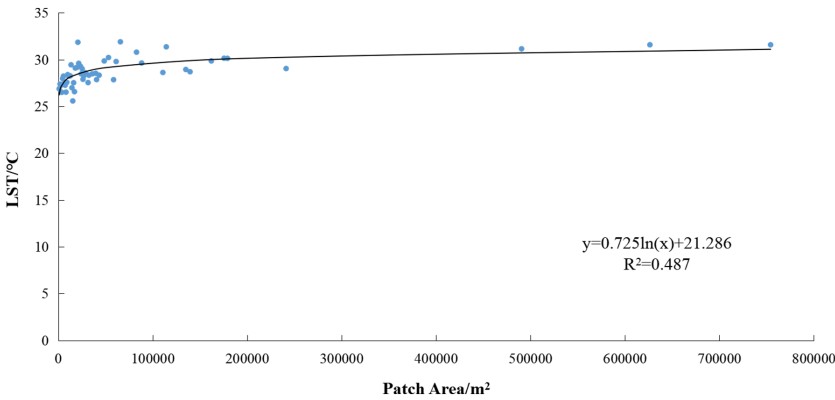

**Figure 8   Correlation between patch area and average patch temperature of mining lands.** LST, land surface temperature.

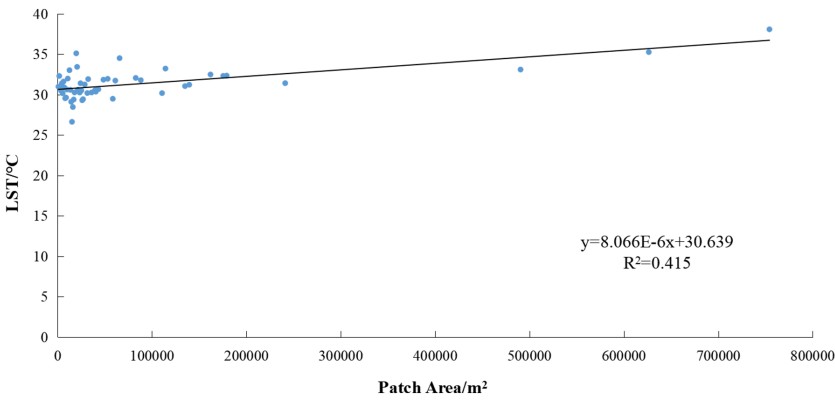

**Figure 9   Correlation between patch area and maximum patch temperature of mining lands.** LST, land surface temperature.

temperature of the mining land. In the 100–200 m buffer zone, the correlation between the temperature and the average area was not significant. Therefore, a higher correlation was found for the entire buffer zone with the average and maximum patch temperatures, while a lower correlation was found with the patch area. Thus, the correlation between the temperature in the buffer zone and the average patch temperature was most relevant.

To further study the correspondence between the average patch temperature of mining land and the temperature in the buffer zones, the 52 data sets were sorted based on their average patch temperatures from smallest to largest. Each of the 13 groups was then compiled into a new group. The average number and the average temperature of the corresponding buffer zone in each new group were calculated to obtain four new data sets (Table 9).

Figure 11 shows that the further the buffer zone was from the mine land patch, the lower its temperature. In 0–200 m buffer zones, the average temperature changed drastically, while the average temperature outside the 200 m zone varied little. The range of this heating

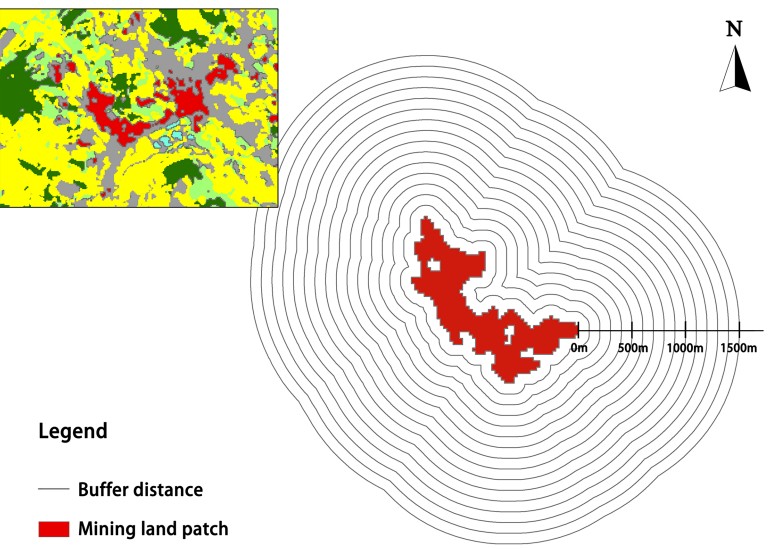

**Legend**

— Buffer distance

■ Mining land patch

**Figure 10  Schematic diagram of buffer zone in mining land patch.** The key has been noted in the figure.

**Table 8  Correlation between LST and buffer zone within the mining lands.**

| Factor | | Average area | Average temperature | Maximum temperature |
|---|---|---|---|---|
| | 100 m | 0.35** | 0.79** | 0.71** |
| | 200 m | 0.07 | 0.41** | 0.39** |
| | 300 m | −0.01 | 0.31* | 0.26 |
| | 400 m | −0.03 | 0.30* | 0.25 |
| | 500 m | 0.01 | 0.33* | 0.30* |
| | 600 m | 0.02 | 0.33* | 0.28* |
| | 700 m | 0.03 | 0.30* | 0.27 |
| Average temperature in buffer zone / °C | 800 m | 0.05 | 0.28* | 0.28* |
| | 900 m | 0.09 | 0.28* | 0.30* |
| | 1,000 m | 0.09 | 0.29* | 0.29* |
| | 1,100 m | 0.09 | 0.29* | 0.28* |
| | 1,200 m | 0.12 | 0.33* | 0.30* |
| | 1,300 m | 0.10 | 0.32* | 0.27 |
| | 1,400 m | 0.08 | 0.29* | 0.24 |
| | 1,500 m | 0.10 | 0.28* | 0.26 |

**Notes.**
*$p < 0.05$.
**$p < 0.01$.

effect is approximately 700 m in Group 1, 1,200 m in Group 2 and 3, and more than 1,400 m in Group 4. Therefore, a larger average patch temperature in the mining land causes a higher temperature in its buffer zone, and the greater the scope of its influence.

**Table 9   Correspondence between LST and buffer zone within the mining lands.**

| Average Temperature/°C | | 27.00 | 28.20 | 29.05 | 30.78 |
|---|---|---|---|---|---|
| | 100 m | 25.15 | 25.84 | 26.01 | 27.17 |
| | 200 m | 24.39 | 24.75 | 24.43 | 25.36 |
| | 300 m | 24.34 | 24.65 | 24.16 | 25.11 |
| | 400 m | 24.22 | 24.64 | 24.07 | 25.00 |
| | 500 m | 24.06 | 24.69 | 24.04 | 24.97 |
| | 600 m | 23.87 | 24.63 | 24.12 | 24.78 |
| Average temperature in differ- | 700 m | 23.76 | 24.53 | 24.14 | 24.59 |
| ent scale buffers /° C | 800 m | 23.70 | 24.44 | 23.95 | 24.52 |
| | 900 m | 23.64 | 24.24 | 23.80 | 24.47 |
| | 1,000 m | 23.59 | 24.06 | 23.77 | 24.41 |
| | 1,100 m | 23.54 | 23.87 | 23.72 | 24.29 |
| | 1,200 m | 23.48 | 23.68 | 23.71 | 24.21 |
| | 1,300 m | 23.43 | 23.57 | 23.68 | 24.07 |
| | 1,400 m | 23.43 | 23.55 | 23.66 | 23.96 |
| | 1,500 m | 23.42 | 23.52 | 23.51 | 23.92 |

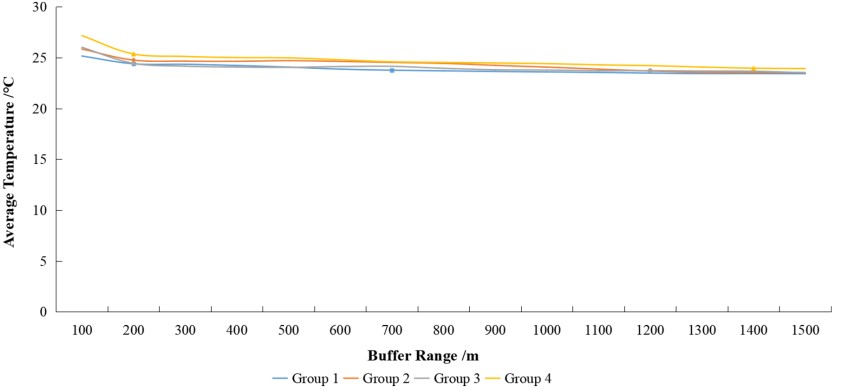

**Figure 11   Variation of land surface temperature (LST) with buffer zone of mining lands.** The double line on the coordinate axis represents the omitted part of the value, so the *Y*-axis can more clearly reflect the trend of the four sets of data in the figure.

## DISCUSSION

### Impact of coal mining activities on surface temperatures

As the largest coal city in Heilongjiang Province, Jixi has always utilized coal as its leading industry. The main types of coal mining wasteland in Jixi City include mining subsidence, land occupation, polluted wasteland, and excavated land, which account for 0.48%, 82.0%, 6.82%, and 10.71% of the total coal mining wasteland, respectively (*Di, Guan & Zheng, 2015*). Coal mining activities generate a significant amount of heat. Thus, regional heating within the city has intensified when coupled with their high-energy consumption and high-heat producing enterprises (*Hu, Zhao & Dong, 2010*). The ongoing economic development

of mining areas has increased both the population density and heat production from urban infrastructure.

The correlation between LST and coal mining activities has resulted in larger mining lands with higher average and maximum patch temperatures. The available literature has shown that the size, shape, number, and boundary properties of these patches affect their energy transmissions. According to landscape ecological theory, the size and shape of these patches also affect their energy accumulation. Likewise, some researchers have recognized that larger patches of construction land have higher degrees of aggregation, more regular shapes, higher LSTs, and more significant heat island effects (*Yu, 2006*; *Fu, 2001*; *Xie, Wang & Fu, 2011*; *Xu et al., 2015*). Some studies have analyzed different types of disturbances at the interior of mining lands, among which dumps, opencast coal pits, and industrial centers have higher contributions to local warming (*Xie, Wang & Fu, 2011*; *Liu, 2016*). Exposed coal and coal gangue easily absorb heat and cause increased temperatures, while piled coal gangue hills are prone to heat and spontaneous combustion (*Hao, 2011*). Therefore, many factors cause high temperatures in mining land.

Quantitative research on the impact of mining land indicates a strong warming effect within a buffer zone of 0–200 m around mining land patches. As the distance from the mining land increases, the warming effects gradually weakens. Mining land patches with higher average patch temperatures have larger temperature-affected buffer zones. Changes in the local meteorological conditions, such as temperature rise, affect local species, which impacts the ecological conditions of the entire region. However, the strength of the warming effect and the size of its influence range are not only related to the distance from the mining land patch but may also be related to the average temperature of the entire area during the analysis (*Liao, 2009*). This specific correlation requires further study.

To date, regulations on the ecological and environmental protection are aimed only at the ecological and environmental indicators within the mining area, which cannot achieve regional ecological protection. Although it seems intuitive that coal production enterprises or units engaged in corresponding activities have taken the responsibility of protecting the ecology and environment, this does not cover the entire affected area of coal mining production activities. To protect the ecological quality of the area while developing coal resources, the scope of environmental protection in mining areas should be defined more scientifically and rationally.

## Impact of different land-use types on surface temperature

Our results show that land-use types have a dominating impact on the LST. The LSTs of the Jiguan, Didao, and Chengzihe District of Jixi were primarily within the range of 16.29–42.29 °C in the two considered years. The low-middle-temperature zone had the largest area, which accounted for 70.53% and 72.21% of the total area. The low-temperature zone was distributed primarily over water areas, forest lands and cultivated lands. The high-temperature zone was distributed mostly over the construction land and mining land, especially the Shenghe Coal Mine in the Didao District and the Chengshan Coal Mine in the Chengzihe District.
The temperatures in 2019 were generally lower than those in 2015. From a normalized comparison, it is seen that the high-temperature and low-temperature zones increased in 2019. Along with the clustered development of mining land patches, the land surface temperature shows a polarizing trend. The expansion of some high-temperature zones may be due to the continued development of coal mines. The increased low-temperature areas may be due to the reclamation and restoration of vegetation in mining areas. Based on governmental planning (''Mineral Resources Planning of Jixi City (2016-2020)'' and ''Special Planning for Reclamation and Utilization of Desert Land of Industrial Mining Area and Mining Subsidence Area in Jixi City (2014–2020)'') from 2015 to 2019, the coal industry wastelands in Chengzihe and Didao Districts were treated to a certain extent, and the reclaimed land was converted into cultivated land, forest land, and construction land. These lands will be used for agricultural production, creating recreational landscapes, and improving the ecological environment.

In recent years, the development of coal resources in Jixi has been rapid. Additionally, the spatial distribution of mines has also changed (*Yang, 2013*). Construction and mining activities have reduced the ''cooling'' land-use types, such as forest and cultivated lands (*Wang et al., 2020*), and replaced them with ''warming'' types, like construction and mining lands. The available literature has shown that urban expansion is the main driving process of land cover changes and consequently rise of LST (*Pal & Ziaul, 2017*), which is consistent with our findings. With changes in land-use types, natural vegetation has been replaced by impervious concrete and construction land, which has caused significant changes like heat radiation from the underlying city surface (*Wang et al., 2013*). These man-made surfaces have a strong light absorptive effect and can quickly raise the local LST (*Hien et al., 2011*). In addition, building facades can reflect light multiple times, heating the near-surface atmosphere and cause LSTs to rise significantly (*Miao et al., 2009*). Some studies have also shown that the heating effect of construction lands, especially compact low-rise buildings, is very obvious (*Das, Das & Mandal, 2020*). Among the six considered land-use types, the LSTs of water area, forest land, and cultivated land were lower than the average LST for the study area. Water-permeable areas of the study region, such as water areas and forest land, ensure efficient heat exchange between the soil and the atmosphere. Water can evaporate, which absorbs heat in the environment and has an overall cooling effect (*Zhang et al., 2013a*; *Zhang et al., 2013b*). Therefore, not only by balancing the land-use types, but also by optimizing appropriate urban planning, the increase in LST can be adjusted to reduce the impact of urbanization on the ecological environment (*Das & Das, 2020*).

## Impact of vegetation coverage on land surface temperatures

Our coupling analysis showed that changes in vegetation coverage are very important factors that affecting ecological status change. There is a significant negative correlation between LST and vegetation coverage, which has also been confirmed by other works (*Estoque, Murayama & Myint, 2017*; *Jiang, Zeng & Zeng, 2015*; *Duan & Zhang, 2012*; *Wu, Xu & Tan, 2007*; *Yue, Xu & Xu, 2006*). As vegetation blocks sunlight, it reduces the amount of solar radiation that reaches the surface, while plant transpiration also reduces the LST (*Cui, Li & Ji, 2018*). In areas with high vegetation coverage, the LST was lower than in

other areas, illustrating the degree to which vegetation could effectively alleviate heat island effect.

Therefore, municipal bodies should carefully consider the balance between ecological protection and economic development. The focus should be on vegetation restoration and environmental governance in areas where heat emissions are concentrated, such as abandoned mine sites and barren areas. Meanwhile, increasing the proportion of green space, improving the diversity and complexity of the landscape, and dividing the impervious surface with vegetation when developing urban construction land and coal mines can significantly reduce the LST and alleviate heat island effects.

## CONCLUSIONS

Our findings show that coal mining activities and urban expansion are the primary factors affecting LSTs. These two factors change land-use types and vegetation coverage, which results in an abnormal heat flux. There were large differences in the LSTs among the various land-use types in Jixi City. The LSTs for the considered land-use types were ranked from high to low, as follows: mining land > construction land > grassland > cultivated land > forest land > water area. The average LST difference between the mining land and water area was more than 10 °C each year.

Correlations between LST and vegetation coverage indicate that they have a significant negative relationship. The LST was lower in areas with higher vegetation coverage than in other areas. For every 0.1 increase in vegetation coverage, the surface temperature is expected to drop by approximately 0.75 °C, indicating the extent to which vegetation can effectively alleviate warming effects.

The correlation between the LST and coal mining activities indicates the patch area of the mining land has a significant positive correlation with both the average and maximum patch temperatures. The average patch temperature shows a logarithmic increase with the growth of the patch area; thus, the average patch temperature increases significantly within 200,000 m². The maximum patch temperature shows a linear increase with the growth of the patch area; thus, the maximum patch temperature increases by approximately 0.81 °C for every 100,000 m² increase in the patch area of mining land. A higher correlation was found between the average patch temperature and the temperature in the buffer zone. This study found that the higher the average patch temperature of mining land, the higher the temperature in its buffer zone, and the greater the scope of its influence. As the distance from the mining land increased, its heating effect weakened.

Full consideration should be given to vegetation restoration in mining areas to reduce the warming effect from coal mining activities, especially in abandoned mining land, by increasing the total vegetation coverage in the study area. The existing large coal mine land patches need to be divided by plants or water areas. Thus, the scope of environmental protection in mining areas needs to be correctly defined. Meanwhile, in future urban layouts, downtown areas should maintain a proper distance from coal mining land. This study provides a useful reference to explore the warming effects caused by coal mining activities and the definition of its influence scope.

### Funding

This work was supported by the Fundamental Research Funds for the Central Universities (No.2572017CA12, No.2572018CP06). The funders had no role in study design, data collection and analysis, decision to publish, or preparation of the manuscript.

### Grant Disclosures

The following grant information was disclosed by the authors:
Fundamental Research Funds for the Central Universities: No.2572017CA12, No.2572018CP06.

### Competing Interests

The authors declare there are no competing interests.

### Author Contributions

- Jia-shuo Cao and Zheng-yu Deng conceived and designed the experiments, performed the experiments, analyzed the data, prepared figures and/or tables, authored or reviewed drafts of the paper, and approved the final draft.
- Wen Li conceived and designed the experiments, analyzed the data, authored or reviewed drafts of the paper, and approved the final draft.
- Yuan-dong Hu analyzed the data, authored or reviewed drafts of the paper, and approved the final draft.

### Data Availability

Data is available at Open Science Framework: Dengzhengyu. 2020. ''Remote Sensing Inversion and Spatial Variation of Land Surface Temperature over Mining Areas of Jixi, Heilongjiang, China.'' OSF. November 3. https://osf.io/tj9wh/.

### Supplemental Information

Supplemental information for this article can be found online at http://dx.doi.org/10.7717/peerj.10257#supplemental-information.

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
