# Peer review of "Remote sensing inversion and spatial variation of land surface temperature over mining areas of Jixi, Heilongjiang, China"

_PeerJ, doi:10.7717/peerj.10257_

## Round 0.1 · original submission · Major Revisions

Please follow the reviewers' comments carefully and revise your manuscript accordingly. Reviewers 3 and 4 have made substantive comments, which include concerns on methodology (outdated and not adequately referenced, as examples) as well as presentation (language and focus).

Reviewer 1 ·

Basic reporting

The introduction has sufficient literature references and reasonable hypothesis.

Experimental design

The experimental design is good.

Validity of the findings

The findings are solid and valid.

Additional comments

This study investigated the spatial variation of Land Surface Temperature over Mining Areas in Southeastern China using Landsat-8 imagery. The results indicated that the mining activities have led to significant increases in LSTs, linked to vegetation loss and environmental degradation. In general, this manuscript is well written and easy to follow, making significant contributions to the literature.

Reviewer 2 ·

Basic reporting

The standardization of spatial thematic mapping should be strengthened. The legend should include all the data-layers.

Experimental design

Methods described with sufficient detail & information to replicate.

Validity of the findings

The conclusions can be further refined, and the tables and charts should be compressed.

Additional comments

Mining area is a kind of complex human activity, its internal type is complex, including mining site and industrial facility site. The mining area with high surface temperature is mainly caused by the operation of factory facilities. The current conclusion generally says that the temperature in the mining area is higher than that in the urban area, and there may be some problems. It is suggested that higher resolution remote sensing images can be used to obtain the data of inner structure or field operation of the mining area, and the reasons of influencing the temperature level should be discussed more pertinently.

Reviewer 3 ·

Basic reporting

The mnuscript is well within the scope of the journal. However, it requires few improvements before being accepted.

Experimental design

The framework and structure of the manuscript is alright.

Validity of the findings

The manuscript presents a good overview over the relationship between LULC and LST.

Additional comments

The present work is very interesting. However, I have observed few basic fundamental issues in the manuscript, which needs to be addressed by the authors before the acceptance of this manuscript. Hence, I recommend major recommendation before being accepted.
Specific Comments to the Authors:
1. The authors have used maximum likelihood classifier for the image classification. It is not clear why the authors have employed such outdated techniques, when more updated and better methods like random forest etc. are available. It has received wide applicability in producing comparatively better LULC.
2. In the methodology section, authors have mentioned about confusion matrix for the validation of LULC. Again, in the result section, they showed the values of Kappa coefficient. In this case, the authors have not provided any clear view on the number of samples that are adopted for validation and how the sample size is determined.
3. In case of vegetation coverage estimation, the authors have mentioned NDVIsoil and NDVIveg. It is not clear how the authors have obtained the NDVI value of pure soil and pure vegetation. There are also no citations against those arguments.

Reviewer 4 ·

Basic reporting

The paper mainly deals with the assessment of spatial variation of Land Surface Temperature (LST) over a Mining area of China using GIS and RS techniques. Most of the methods used in this study are backdated and these were used widely used in many previous research studies. However, a number of serious issues preclude me from positively recommending the paper for publication.
1. The introduction is too short to raise the scientific contribution of the study.
2. The introduction fails to address the novelty of the work. Author should properly address the previous literatures dealing with the assessment LST over the mining cities.
3. Language needs to be changed.
4. Line 40, author states “In the process of resource development in resource-based cities, the land-use patterns in mining areas are constantly changing, which brings about a series of impacts on the regional 42 ecological environment” without any valid scientific reference.
5. Table 1 is unnecessary. So it would better to remove the Table 1
6. In Table 2 and 4, author calculated normalized temperature index but how they have identified the threshold value for each class?
7. In Table 3 and6, I hope It would be better to add; Standard Deviation’.
8. Authors analyzed buffer zone but not showed on maps.

Experimental design

1. The authors just considered year 2019 for showing the spatial vitiation of LST over the study landscapes. But without temporal analysis, it has no scientific base. If one never analyzes the temporal pattern of LST then how one can know whether the LST has changed or not even how fat different geophysical parameters have a role on LST or over the study area?
2. The author should change the discussion part. In discussion part author gave more focus on the result of the study. In discussion part author must deal the rationality of the result of the other studies.
3. Teh reserch questions of the study needs to be addressed properly
5. The methods needs to be more scientific and updated

Validity of the findings

1. The heading of the discussion part creates ambiguities. Author should correctly write the subheading in discussion part.
2. Author must address the major driers of spatial variation of LST of the study area.
3. The findings of the result were not properly adressed.
4. The athors should clearly focus the litertures of the pervious reaserch study and relevent litertures needs to be used.

---

## Round 0.2 · Minor Revisions

Both reviewers have additional minor comments on your manuscript for you to consider. One reviewer has suggested adding some recent references; if you feel the ones provided are appropriate, please do so.

Reviewer 3 ·

Basic reporting

The authors have used Maximum likelihood classifier for the image classification. It is not clear why the authors have employed such outdated techniques when more updated and better methods like random forest etc. are available. It has received wide applicability in producing comparatively better LULC.

Experimental design

no comment

Validity of the findings

no comment

Reviewer 4 ·

Basic reporting

Although author revised the paper as per as suggestions but still there is a lack of important literatures . Therefore I would suggest author to use following references for enrichment of the papers
(i) Das, M., & Das, A. (2020). Assessing the relationship between local climatic zones (LCZs) and land surface temperature (LST)–A case study of Sriniketan-Santiniketan Planning Area (SSPA), West Bengal, India. Urban Climate, 32, 100591. https://doi.org/10.1016/j.uclim.2020.100591
(2) Das, M., Das, A., & Mandal, S. (2020). Outdoor thermal comfort in different settings of a tropical planning region of Eastern India by adopting LCZs approach: A case study on Sriniketan-Santiniketan Planning Area (SSPA). Sustainable Cities and Society, 102433.https://doi.org/10.1016/j.scs.2020.102433
(3) Estoque, R. C., Murayama, Y., & Myint, S. W. (2017). Effects of landscape composition and pattern on land surface temperature: An urban heat island study in the megacities of Southeast Asia. Science of the Total Environment, 577, 349-359.
(4) Pal, S., & Ziaul, S. K. (2017). Detection of land use and land cover change and land surface temperature in English Bazar urban centre. The Egyptian Journal of Remote Sensing and Space Science, 20(1), 125-145.
(6) Wang, R., Hou, H., Murayama, Y., & Derdouri, A. (2020). Spatiotemporal Analysis of Land Use/Cover Patterns and Their Relationship with Land Surface Temperature in Nanjing, China. Remote Sensing, 12(3), 440.

Experimental design

The contents are very suitable for this journal. But it has few common mistakes. I can't identity the research gaps. Please identify and specify

Validity of the findings

I would suggest author to add above mentioned literature to stand the novelty of the paper. Without this, I can't recommend to accept the paper

Additional comments

Although the authors have made a good attempt to all the responses. But still i found some mistakes.
(i) In figure 2 red color denotes 'high' but in figure 5 it denotes 'low'. Please use same color scheme. Even use common knowledge in using color scheme. When you are using red color it never treats as low, It has its own significance.
(ii) In line 161, it should be 'Where' not 'where'. Please check all the mistakes.

---

## Round 0.3 · accepted · Accept

Thank you for your efforts in revising your manuscript.